# Elaborating the phase diagram of spin-1 anyonic chains

Eric Vernier[1*], Jesper Lykke Jacobsen[2,3,4] and Hubert Saleur[4,5]

**1** SISSA and INFN, Sezione di Trieste, via Bonomea 265, I-34136, Trieste, Italy
**2** LPTENS, École Normale Supérieure – PSL Research University, 24 rue Lhomond,
F-75231 Paris Cedex 05, France
**3** Sorbonne Universités, UPMC Université Paris 6, CNRS UMR 8549,
F-75005 Paris, France
**4** Institut de Physique Théorique, CEA Saclay, F-91191 Gif-sur-Yvette, France
**5** USC Physics Department, Los Angeles CA 90089, USA

\* evernier@sissa.it

## Abstract

We revisit the phase diagram of spin-1 $su(2)_k$ anyonic chains, originally studied by Gils *et. al.* [9].These chains possess several integrable points, which were overlooked (or only briefly considered) so far.

Exploiting integrability through a combination of algebraic techniques and exact Bethe ansatz results, we establish in particular the presence of new first order phase transitions, a new critical point described by a $Z_k$ parafermionic CFT, and of even more phases than originally conjectured. Our results leave room for yet more progress in the understanding of spin-1 anyonic chains.



# 1   Introduction

Non-Abelian anyons are quasiparticles with exotic braiding statistics which have been proposed to occur as emergent low-energy excitations in certain 2+1-dimensional quantum materials. In particular, their presence has been theoretically predicted in $p_x + ip_y$ superconductors [1] as well as in certain fractional quantum Hall states [2]. Over the last years, there has been some experimental evidence for non-Abelian edge states (the celebrated Majorana fermions) in nanowires [3, 4]. Importantly, non-abelian anyons may form collective states with exponentially precise degeneracies that could be used for topological quantum computation [5].

In this context it is of crucial importance to understand how such particles interact, and over the years a lot of attention has been given to one-dimensional anyonic chains with short-range interactions. These may be viewed as generalizations of the more usual quantum spin chains appearing in the description of quantum magnets. In particular, interacting chains of the so-called $su(2)_k$ anyons (such as the Ising and Fibonacci anyons, corresponding to $k = 2$ and $k = 3$ respectively) [6–9] are generalizations of the $su(2)$ Heisenberg spin chains, the latter being recovered in the $k \to \infty$ limit. Other chains based on deformations of different classical groups have also been studied [10–15]. While the group $su(2)$ allows for representations of arbitrarily large integer or half-integer angular momenta, in $su(2)_k$ these representations are restricted to $j = 0, \frac{1}{2}, \ldots, \frac{k}{2}$; the fusion rules are accordingly deformed, resulting in non-trivial braiding relations. The first example studied in some detail was that of a chain of spin-1/2 $su(2)_3$ anyons, the so-called 'golden chain' [7]. This was followed by a generalization to arbitrary $k$, and an extension to spin-1 chains [8,9]. In both cases the phase diagrams were shown to have strong resemblances with that of the usual Heisenberg spin chains, namely, in the spin-1/2 case a gapless ground state described by conformal field theory, and in the spin-1 case a gapped topological phase generalizing the Haldane phase to anyons [16, 17]. Along with this, several new features were also exhibited, in particular the spin-1 phase diagram was argued in [8,9] to possess a rich variety of gapped and gapless phases, as well as an intriguing dependence on the parity of the parameter $k$. Despite an extensive analysis based on exact diagonalization and conformal field theory arguments, several aspects of the phase diagram remained mysterious.

We point out in this note that the phase diagram of the spin-1 anyonic chain possesses several integrable points which were overlooked or underutilized in [9]. By a thorough analysis of the behavior of the chain at these points, we are able to complement—and in several instances, profoundly correct—the results in [9].

The plan of the paper is as follows. In section 2, we introduce the spin-1 $su(2)_k$ anyonic chains. In section 3, we consider the simplest Temperley-Lieb points. We use this opportunity

to discuss in detail the obstacles in relating physical properties of models which are based on the same algebra, but with different representations (such as vertex, loop, and anyonic or restricted solid on solid (RSOS) models). We remind the reader of the (not so well known) results about integrability in spin-1 systems in 4, and quickly discuss the Fateev-Zamolodchikov points. Most new results concern the new, Izergin-Korepin points, and are discussed in section 5. The consequences for the phase diagram of our findings are discussed in section 6.

## 2 The model

We refer to [18, 19] for a general introduction to the properties of anyons. The model we consider was introduced in references [8, 9] to which we also refer for an extensive discussion of $su(2)_k$. In oder to facilitate comparison with [9], we will use the same notations whenever possible. The properties of $su(2)_k$, where $k \geq 2$ is an integer, can be understood as a generalization of those of $SU(2)$, in that particles belong to representations of fixed integer of half-integer 'angular momentum' $j$, and combine through 'fusion rules'. In contrast with $su(2)$, however, in $su(2)_k$ the set of allowed representations is truncated to

$$j = 0, \frac{1}{2}, \ldots, \frac{k}{2},$$

and the fusion rules take the form

$$j_1 \times j_2 = |j_1 - j_2|, |j_1 - j_2| + 1, \ldots, \min(j_1 + j_2, k - j_1 - j_2).$$

In this work we consider a chain of $L$ spin-1 anyons, with nearest-neighbour interactions and periodic boundary conditions. For $k \geq 4$, to which we will restrict in the present work, each pair of neighbouring anyons can be fused in either of the three $j = 0, 1, 2$ channels resulting from the fusion $1 \times 1 = 0 + 1 + 2$, and the most general Hamiltonian with nearest-neigbour interaction can be defined by assigning a different energy to each of these fusion channels. Namely, it can be written (up to some additive identity term and some overall multiplication factor) as

$$H = \sum_{i=0}^{L-1} \cos \theta_{2,1} P_i^{(2)} - \sin \theta_{2,1} P_i^{(1)}, \tag{1}$$

where the operator $P_i^{(l)}$ projects onto the channel $l = 0, 1, 2$ of the fusion $1 \times 1 = 0 + 1 + 2$ involving the $i^{\text{th}}$ and $i + 1^{\text{th}}$ particles. We have here used the relation $P_i^{(0)} + P_i^{(1)} + P_i^{(2)} = \text{id}$. For completeness, we recall the general expression of the projectors [9, equation (C1)]:

$$P_i^{(l)}|x_0, \ldots, x_{i-1}, x_i, x_{i+1}, \ldots x_{L-1}\rangle =$$
$$\sum_{x_i'} \left(F_{x_{i+1}}^{x_{i-1},j,j}\right)_{x_i}^l \left(F_{x_{i+1}}^{x_{i-1},j,j}\right)_{x_i'}^l |x_0, \ldots, x_{i-1}, x_i', x_{i+1}, \ldots, x_{L-1}\rangle, \tag{2}$$

for spin $j$ anyons and the fusion channel $l$, and where the $x_i = 0, \ldots, \frac{j}{2}$. Throughout most of the paper we will use periodic boundary conditions, $x_0 = x_L$. For fully explicit forms of this Hamiltonian for several values of $k$ we refer to [9, appendix C]. Let us mention before proceeding that for $k = 2, 3$, the fusion relation $1 \times 1$ instead closes onto the representations $j = 0, 1$, so a Hamiltonian such as (1) may not be defined. In addition, for $k = 4$, the model (1) is integrable for any value of the angle $\theta_{2,1}$ and related to the Ashkin-Teller class [15, 20].

As noted in [9], in the $k \to \infty$ limit (1) becomes equivalent to the $SU(2)$ spin-1 bilinear-biquadratic Hamiltonian

$$H = \sum_i \cos \theta_{bb} \left(\vec{S}_i \cdot \vec{S}_{i+1}\right) + \sin \theta_{bb} \left(\vec{S}_i \cdot \vec{S}_{i+1}\right)^2,$$

where the relation between $\theta_{bb}$ and $\theta_{2,1}$ is given by

$$\tan\theta_{bb} = \frac{\tan\theta_{2,1} + 1/3}{1 + \tan\theta_{2,1}}\,.$$

The phase diagram of (1) as a function of the parameter $\theta_{2,1}$ has been studied in [9], mostly using numerical techniques. We recall in figure 1 the main results.

## 3 Integrability and representations: the Temperley-Lieb points

### 3.1 Integrability and representations

As already noted in [9], there are several points in the phase diagram of the model (1) where integrability techniques could be used to obtain the properties exactly, or at least with considerable accuracy using numerical solutions of the Bethe ansatz equations.

This is, however, a delicate exercise. Integrability—here the fact that the Hamiltonian belongs to an infinite family of commuting transfer matrices— is usually based on the presence of an underlying lattice algebra such as the Temperley-Lieb algebra or the Birman-Murakami-Wenzl algebra. The Temperley-Lieb algebra, for instance, has been encountered early on in the study of anyonic spin-1/2 chains [7, 8]. In this case, since the product of two spins $1/2$ decomposes on only two channels, $\frac{1}{2} \times \frac{1}{2} = 0 + 1$, and since $P_i^{(0)} + P_i^{(1)} = \text{id}$, the most general nearest neighbor Hamiltonian is proportional to $P_i^{(0)}$, and it is well known that this object (or rather, $E_i = d_{1/2} P_i^{(0)}$, with $d_{1/2} = 2\cos\frac{\pi}{k+2}$) obeys the Temperley-Lieb relations

$$
\begin{aligned}
{E_i}^2 &= nE_i \\
E_i E_{i\pm 1} E_i &= E_i \\
E_j E_i &= E_i E_j \quad \text{for } |i-j| \geq 2\,,
\end{aligned}
\tag{3}
$$

with $n \equiv d_{1/2}$. Having relations such as (3) allows one to build commuting transfer matrices after proper introduction of a spectral parameter, and to prove, for instance, that the Hamiltonians $H \propto \sum E_i$ are integrable.

Now, lattice algebras admit different representations: loop models (where the degrees of freedom are geometrical objects), vertex or spin models (where the degrees of freedom are $U_q sl(2)$ spin states), and solid-on-solid height models (where the degrees of freedom are 'heights' taking values on certain diagrams). Anyonic systems can also be thought of, in integrable parlance, as restricted solid-on-solid (RSOS) height models. The values $j = 0, \ldots, \frac{k}{2}$ can be considered as the heights of a solid-on-solid interface, and the fact that $j$ is bounded means the model is restricted. See e.g. [7] for more detail on this reformulation of the spin-1/2 anyonic chain.

The crucial point is that *the physics may depend on the representation*. While all representations provide integrable models, it is perfectly possible, for a given form of the Hamiltonian expressed in terms of the algebra, that these different incarnations have different ground states and excited states, which will all form different subsets of the (very large) set of Bethe ansatz states. A well-known example of this is provided by the so called 'transition between regimes I and II' in the Andrews-Baxter-Forrester RSOS models: the ground state of these models sits infinitely high above[1] the ground state of the corresponding vertex model, and their critical properties are totally unrelated. When $n = \sqrt{2}$ (in (3)) for this transition point, the RSOS model is the usual Ising model with central charge $c = \frac{1}{2}$, but the vertex model, which sits on

---

[1]To be precise, the energy per spin of the two ground states are different.

**Sci**|Post                                                          SciPost Phys. 2, 004 (2017)

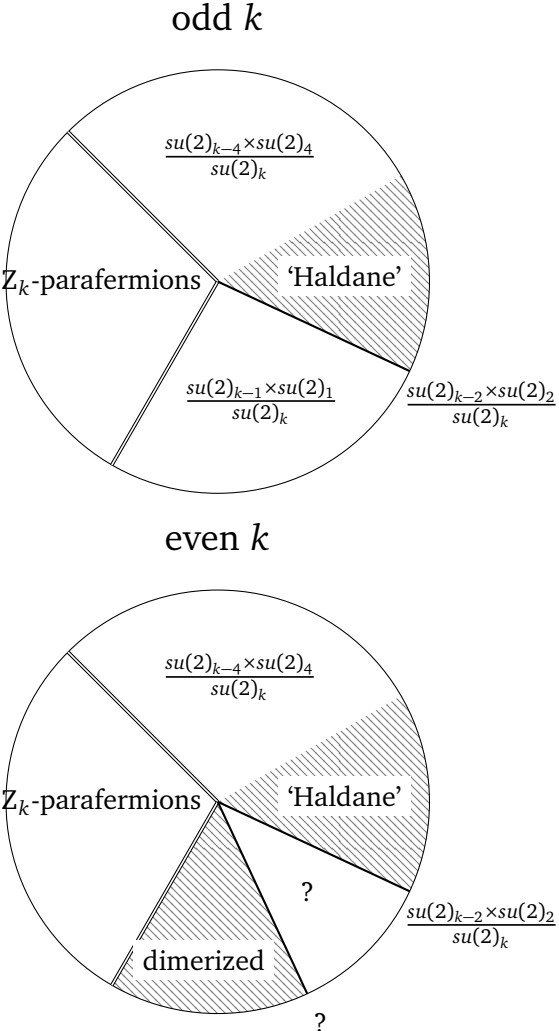

Figure 1: Phase diagram of anyonic spin chains proposed in [9] for $k > 4$ odd (left) and even (right). The nature of the different phases is explained in detail in that reference. We have denoted by doubled lines first order phase transitions. The 'Haldane' and dimerized phases, hashed on the figure, are massive. In the case of $k$ even, the extended region denoted by a question mark is referred to in [9] as an 'extended critical phase', whose precise nature was left unelucidated. The nature of the transition between this phase and the dimerized phase was also not elucidated in [9], and we denote this in the figure by another question mark. All the other regions are supposed to be gapless, and described at low energy by the CFTs indicated on the figure.

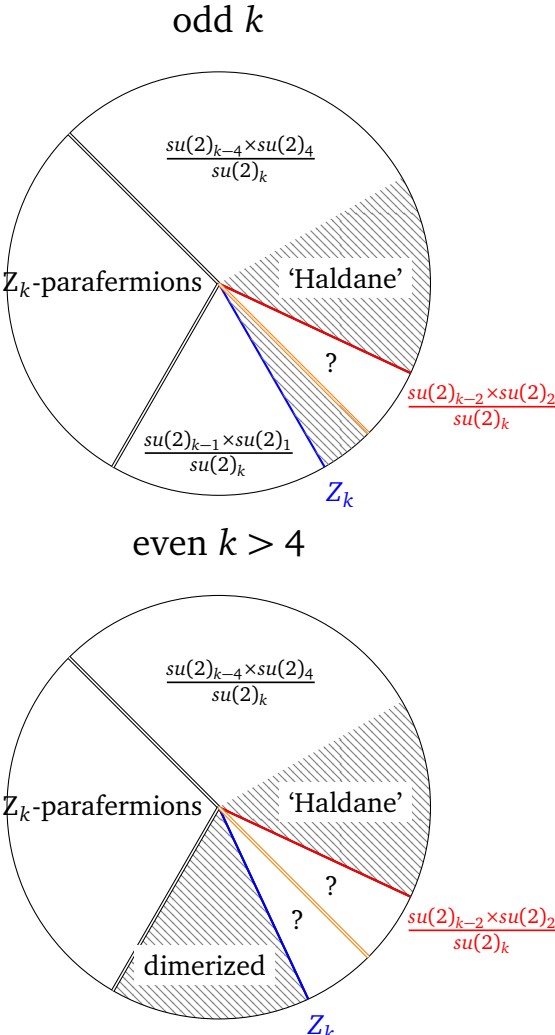

Figure 2: Improved phase diagram resulting from our study. The points $\theta_{2,1} = \theta_{\mathrm{IK}}, \theta_{\mathrm{FZ}}, \theta_{\mathrm{TL}}$ are represented in blue, red and orange respectively. For $k > 4$ the point $\theta_{\mathrm{TL}}$ is not critical, and corresponds to a first order transition. The point $\theta_{\mathrm{IK}}$ is described by a parafermionic $Z_k$ CFT, of a different nature than the extended $Z_k$ phase. For $k$ odd we conjecture that it should be the locus of a transition between the critical $\frac{su(2)_{k-2} \times su(2)_2}{su(2)_k}$ phase and a new massive phase, denoted by hashed lines. We don't know what happens for $k > 4$ even. For $k = 4$ the points $\theta_{\mathrm{IK}}, \theta_{\mathrm{FZ}}, \theta_{\mathrm{TL}}$ coincide, and our results agree with the propositions of [9].

the 'unphysical self-dual line' has $c = -\frac{22}{5}$ ! For a more detailed discussion of this phenomenon see [21–23]. It is this subtlety that prevented the authors of [9] from using, without further work, old results [24] about the behavior of the $U_q sl(2)$ symmetric spin one chain.[2]

The technical effort required to make progress on the issue is however quite reasonable, and only requires the control of simple aspects of representation theory. This is (part of) what we do in this paper.

## 3.2 The Temperley-Lieb points

We start our study with the two following points

$$\theta_{2,1} = \theta_{\text{TL}} \equiv -\frac{\pi}{4}, \quad \text{and} \quad \theta_{2,1} = \pi + \theta_{\text{TL}} = \frac{3\pi}{4}.$$

What is special about these points is that the two projectors in the Hamiltonian (1) come with the *same coefficient*. It follows that the interaction term is proportional to $P_i^{(1)} + P_i^{(2)} = \text{id} - P_i^{(0)}$, and thus the nearest neighbor interaction becomes, up to the trivial contribution of the identity channel, proportional to a single projector $P^{(0)}$. We rewrite in this case the Hamiltonian as

$$H_{\text{TL}} = \mp \sum_{i=1}^{L} E_i \,, \tag{4}$$

where we have defined the operators

$$E_i = \frac{\sin \frac{3\pi}{k+2}}{\sin \frac{\pi}{k+2}} P_i^{(0)} = \frac{\mathfrak{q}^3 - \mathfrak{q}^{-3}}{\mathfrak{q} - \mathfrak{q}^{-1}} P_i^{(0)} = (3)_{\mathfrak{q}} P_i^{(0)} \equiv d_1 P_i^{(0)} \,.$$

In the above equation we have introduced the parameter $\mathfrak{q} = e^{i \frac{\pi}{k+2}}$, as well as the usual notation for $\mathfrak{q}$-deformed numbers,

$$(x)_{\mathfrak{q}} = \frac{\mathfrak{q}^x - \mathfrak{q}^{-x}}{\mathfrak{q} - \mathfrak{q}^{-1}} = \frac{\sin \frac{x\pi}{k+2}}{\sin \frac{\pi}{k+2}} \,,$$

so $d_{1/2} = 2 \cos \frac{\pi}{k+2} = (2)_{\mathfrak{q}}$, and $d_1 = 1 + 2 \cos \frac{2\pi}{k+2} = (3)_{\mathfrak{q}}$. Remarkably, the operators $\{E_i\}_{i=1,\dots,L}$ obey the relations (3), that is, they provide a representation of the Temperley-Lieb algebra, just like the projector onto the singlet in the spin-1/2 case [7]: the difference with the spin-1/2 case is that now the parameter $n$ takes the value[3] $n \equiv d_1$. Models with Hamiltonians (4) have been studied for many years. The question is, can we infer, from all the available results, the physics for our particular anyonic representation?

In order to do so, the proper way to proceed is to get knowledge of all the possible irreducible modules of the algebra, and see which modules arise in which particular representation —that is, physical model. An important fact, which must be emphasized, is that the modules determine fully the spectrum of the Hamiltonian, since the latter is part of the algebra. If two

---

[2]To quote the authors of [9]: "Despite the similarities between the model of Koo and Saleur and our anyonic model, they behave rather differently. The phase boundaries between the various phases observed in the Koo-Saleur model depend smoothly on the continuous parameter $Q$, while the phase diagrams of the anyonic spin 1 models depend on whether $k$ is even or odd. In addition, the Koo-Saleur model displays non-unitary critical behavior, while the critical behavior of the anyon models is described by unitary CFTs. The explanation for this difference in behavior should be sought in the representations used in the two models. In the Koo-Saleur model, a representation which essentially behaves like a $SU(2)$ representation is used (which permits to define the model as a function of the continuous parameter). In the anyonic version, the truncated $su(2)_k$ representations play a central role."

[3]It is well-known that the projector onto the singlet in the tensor product of two spin-$j$ representations obeys (after multiplication by $d_j = (2j+1)_{\mathfrak{q}}$) the Temperley-Lieb relations with parameter $n \equiv d_j$.

Temperley-Lieb Hamiltonians for different models (that is, different representations of the algebra) involve some identical modules, the corresponding part of the spectrum will be the same in both models. Note that this does not preclude the ground state of both models from being different, since they may arise in different modules.

A convenient way to express knowledge of the modules is to use a particular representation of the Temperley-Lieb algebra based on the 6-vertex model. This model has Hamiltonian (4) again, but with a different, non-anyonic Hilbert space, which is now the tensor product of spin-1/2 representations of $U_q sl(2)$, where q is a continuous parameter. In a two-dimensional, statistical version [7] where the Hamiltonian is replaced by a transfer matrix, the states on every edge are labelled by the third component of the spin $s_z = \pm 1/2$, and can be represented by arrows that meet at vertices—hence the name vertex model. Observe now that, since we are interested ultimately in a *periodic anyonic chain*, we must, strictly speaking, consider the periodicized version of the Temperley-Lieb algebra, where relations (3) are 'wrapped up' by setting $E_L \equiv E_0$.

In our case, we have $n = (3)_q$, and we parametrize

$$n = (3)_q = (2)_{q'},$$

that is,

$$1 + q^2 + q^{-2} = q' + q'^{-1}.$$

When $q'$ is a root of unity, the Temperley-Lieb algebra becomes non semi-simple, and extra care has to be taken in analyzing related physical models: this is the case for spin-1/2 anyons. However, in the problem of spin-1 anyons, things are actually much simpler: it is $q = e^{i\pi/k+2}$ which is a root of unity, and, for $k \geq 2$ integer, this corresponds to $q'$ *not a root of unity*. In the 6-vertex model language, and for generic values $n$ of the parameter, the irreducible modules are then classified by two numbers: one is the value of the total spin $\sigma_z$ (it is enough to consider $\sigma_z \geq 0$, and we denote it then by $j$. Note that this is the spin in the equivalent 6 vertex model, not the spin in the spin one chain) and the other is the 'twist angle': even if the algebra is periodic, it can be realized with a vertex model (or spin chain) that is not, and involves a twist $\varphi$ between the first and last spins[4] [25–27]. It turns out that for the anyonic chain at the TL point, for $j \neq 0$, only modules with $\varphi = \frac{2m\pi}{j}$ ($m$ integer) appear. We denote these modules as $\mathcal{V}_{j,\varphi}$. When $j = 0$, another family of modules appears, corresponding to *imaginary* values of the phase:

$$e^{i\varphi/2} + e^{-i\varphi/2} = 1 + 2\cos\frac{2p\pi}{k+2}. \tag{5}$$

We will denote the corresponding modules by $\mathcal{V}_{0,p}$, stressing that in this case the second index does not have the same meaning as for the $\mathcal{V}_{j,\varphi}$ modules.

To proceed, we write a few examples of the decomposition of the spin-1 anyonic chain Hilbert space (denoted in the following as $\mathcal{H}$) in terms of irreducible modules. It will be convenient to separate the study between even and odd values of $L$. For $L$ even, we find:

---

[4]In [25] the twist term $e^{i\varphi}$ appears in the XXZ Hamiltonian as a term coupling the first and last spin, $e^{i\phi}\sigma_L^+\sigma_1^- + \text{h.c.}$

$k = 4$    $L = 4$    $\mathcal{H} = \mathcal{V}_{0,1} \oplus \mathcal{V}_{0,2} \oplus \mathcal{V}_{0,3}$    $\ominus \mathcal{V}_{1,0} \oplus \mathcal{V}_{2,0}$    $\oplus 2\mathcal{V}_{2,\pi}$

        $L = 6$    $\mathcal{H} = \mathcal{V}_{0,1} \oplus \mathcal{V}_{0,2} \oplus \mathcal{V}_{0,3}$    $\ominus \mathcal{V}_{1,0} \oplus \mathcal{V}_{2,0}$    $\oplus 2\mathcal{V}_{2,\pi}$      $\oplus 2\mathcal{V}_{3,\frac{2\pi}{3}}$

$k = 5$    $L = 4$    $\mathcal{H} = \mathcal{V}_{0,1} \oplus \mathcal{V}_{0,2} \oplus \mathcal{V}_{0,3}$    $\oplus 4\mathcal{V}_{2,0} \oplus 4\mathcal{V}_{3,\pi}$

        $L = 6$    $\mathcal{H} = \mathcal{V}_{0,1} \oplus \mathcal{V}_{0,2} \oplus \mathcal{V}_{0,3}$    $\oplus 4\mathcal{V}_{2,0} \oplus 4\mathcal{V}_{2,\pi} \oplus 7\mathcal{V}_{3,0} \oplus 14\mathcal{V}_{3,\frac{2\pi}{3}}$

$k = 6$    $L = 4$    $\mathcal{H} = \mathcal{V}_{0,1} \oplus \mathcal{V}_{0,2} \oplus \mathcal{V}_{0,3} \oplus \mathcal{V}_{0,4}$    $\oplus 6\mathcal{V}_{2,0} \oplus 6\mathcal{V}_{2,\pi}$

        $L = 6$    $\mathcal{H} = \mathcal{V}_{0,1} \oplus \mathcal{V}_{0,2} \oplus \mathcal{V}_{0,3} \oplus \mathcal{V}_{0,4}$    $\oplus 6\mathcal{V}_{2,0} \oplus 6\mathcal{V}_{2,\pi} \oplus 16\mathcal{V}_{3,0} \oplus 32\mathcal{V}_{3,\frac{2\pi}{3}}$

$k = 7$    $L = 4$    $\mathcal{H} = \mathcal{V}_{0,1} \oplus \mathcal{V}_{0,2} \oplus \mathcal{V}_{0,3} \oplus \mathcal{V}_{0,4}$    $\oplus \mathcal{V}_{1,0} \oplus 9\mathcal{V}_{2,0} \oplus 8\mathcal{V}_{2,\pi}$

        $L = 6$    $\mathcal{H} = \mathcal{V}_{0,1} \oplus \mathcal{V}_{0,2} \oplus \mathcal{V}_{0,3} \oplus \mathcal{V}_{0,4}$    $\oplus \mathcal{V}_{1,0} \oplus 9\mathcal{V}_{2,0} \oplus 8\mathcal{V}_{2,\pi} \oplus 25\mathcal{V}_{2,0} \oplus 48\mathcal{V}_{3,\frac{2\pi}{3}}$

$k = 8$    $L = 4$    $\mathcal{H} = \mathcal{V}_{0,1} \oplus \mathcal{V}_{0,2} \oplus \mathcal{V}_{0,3} \oplus \mathcal{V}_{0,4} \oplus \mathcal{V}_{0,5} \oplus \mathcal{V}_{2,0} \oplus 11\mathcal{V}_{2,0} \oplus 10\mathcal{V}_{2,\pi}$

        $L = 6$    $\mathcal{H} = \mathcal{V}_{0,1} \oplus \mathcal{V}_{0,2} \oplus \mathcal{V}_{0,3} \oplus \mathcal{V}_{0,4} \oplus \mathcal{V}_{0,5} \oplus \mathcal{V}_{1,0} \oplus 11\mathcal{V}_{2,0} \oplus 10\mathcal{V}_{2,\pi} \oplus 34\mathcal{V}_{3,0} \oplus 66\mathcal{V}_{3,\frac{2\pi}{3}}$

Several comments are in order. First, for a given $L$ the action of all TL generators in sectors with $2j = L$ is zero, irrespectively of the corresponding twist $\varphi$, and the modules associated with different values of $\varphi$ are in principle indistinguishable. The reason for which we are able to write, for instance, $\mathcal{V}_{4,0} \oplus 2\mathcal{V}_{4,\pi}$ at $L = 4, k = 4$, comes from the observation at larger sizes that this decomposition is indeed the one which holds. Second, the presence of minus signs ("$\ominus$") in the decompositions for $k = 4$ signals the fact that in this case $\mathcal{V}_{2,0}$ is contained in $\mathcal{V}_{0,1}$, in other terms the representations are not simple (this happens, even for $q'$ generic, for special values of $\varphi$).

In the case of $L$ odd we find

$k = 4$    $L = 3$    $\mathcal{H} = \mathcal{V}_{1/2,\pi} \oplus 2\mathcal{V}_{3/2,\pi} \oplus 2\mathcal{V}_{3/2,\frac{\pi}{3}}$

        $L = 5$    $\mathcal{H} = \mathcal{V}_{1/2,\pi} \oplus 2\mathcal{V}_{3/2,\pi} \oplus 2\mathcal{V}_{3/2,\frac{\pi}{3}} \oplus \mathcal{V}_{5/2,\pi}$

$k = 5$    $L = 3$    $\mathcal{H} = 2\mathcal{V}_{1/2,\pi} \oplus 3\mathcal{V}_{3/2,\pi} \oplus 2\mathcal{V}_{3/2,\frac{\pi}{3}}$

        $L = 5$    $\mathcal{H} = 2\mathcal{V}_{1,\pi} \oplus 3\mathcal{V}_{3/2,\pi} \oplus 2\mathcal{V}_{3/2,\frac{\pi}{3}} \oplus 4\mathcal{V}_{5/2,\pi} \oplus 4\mathcal{V}_{5/2,\frac{\pi}{5}} \oplus 4\mathcal{V}_{5/2,\frac{2\pi}{5}}$

$k = 6$    $L = 3$    $\mathcal{H} = 2\mathcal{V}_{1,\pi} \oplus 4\mathcal{V}_{3/2,\pi} \oplus 4\mathcal{V}_{3/2,\frac{\pi}{3}}$

        $L = 5$    $\mathcal{H} = 2\mathcal{V}_{1,\pi} \oplus 4\mathcal{V}_{3/2,\pi} \oplus 4\mathcal{V}_{3/2,\frac{\pi}{3}} \oplus 6\mathcal{V}_{5/2,\pi} \oplus 8\mathcal{V}_{5/2,\frac{\pi}{5}} \oplus 8\mathcal{V}_{5/2,\frac{2\pi}{5}}$

$k = 7$    $L = 3$    $\mathcal{H} = 3\mathcal{V}_{1,\pi} \oplus 5\mathcal{V}_{3/2,\pi} \oplus 4\mathcal{V}_{3/2,\frac{\pi}{3}}$

        $L = 5$    $\mathcal{H} = 3\mathcal{V}_{1,\pi} \oplus 5\mathcal{V}_{3/2,\pi} \oplus 4\mathcal{V}_{3/2,\frac{\pi}{3}} \oplus 9\mathcal{V}_{5/2,\pi} \oplus 12\mathcal{V}_{5/2,\frac{\pi}{5}} \oplus 12\mathcal{V}_{5/2,\frac{2\pi}{5}}$

and similar comments as in the even case hold regarding the degeneracy between modules $\mathcal{V}_{j=L,\varphi}$ for a given size $L$.

It is useful to compare the above decompositions with the decomposition for the $Q$-state Potts model. This well-known model is defined initially for $Q$ integer only, but its definition can be extended to arbitrary values of $Q$ by using a high-temperature expansion in terms of clusters of identical spins—the so-called Fortuin-Kasteleyn representation. The correspondence between $Q$ and $n$ is simply $Q = n^2$. It is thus natural to expect that our anyonic model at the Temperley-Lieb points is related in some sense with the $Q = (3)_q^2$ state Potts model. The two signs of the Hamiltonian correspond respectively to the critical self-dual Potts model, and to the self-dual Potts model on its "non-physical" critical self-dual line [21].

In the Potts model, the anyonic level $k$ is replaced by the continuous variable $Q$, and the spin $j \equiv \ell$ in the modules can be interpreted geometrically as the number of distinct non-contractible clusters that are preserved by the time evolution. The decomposition of the Hilbert space then reads

$$\mathcal{H} = \mathcal{V}_{0,1} \oplus (Q-1)\mathcal{V}_{1,0} \oplus \bigoplus_{\ell=2}^{L} \bigoplus_{\substack{m=1 \\ m|\ell}}^{\ell} \alpha_{\ell,m}(Q)\mathcal{V}_{\ell,\varphi=\frac{2\pi m}{\ell} \bmod 2\pi}, \tag{6}$$

where, in the last sum, $m$ is a divisor of $\ell$. The $Q$-dependent multiplicities are given by [28–30]

$$\alpha_{\ell,m}(Q) = \Lambda_{\ell,m,Q} + \frac{Q-1}{2}\Lambda_{\ell,m,0}, \tag{7}$$

where

$$\Lambda_{\ell,m,Q} = \frac{2}{\ell} \sum_{d>0:d|\ell} \frac{\mu\left(\frac{m}{m\wedge d}\right)\phi\left(\frac{\ell}{d}\right)}{\phi\left(\frac{m}{m\wedge d}\right)} T_{2d}\left(\tfrac{1}{2}\sqrt{Q}\right). \tag{8}$$

Here $a \wedge b \equiv \gcd(a, b)$ denotes the greatest common divisor. The Möbius function $\mu$ is defined by $\mu(n) = (-1)^r$, if $n$ is an integer that is the product $n = \prod_{i=1}^r p_i$ of $r$ *distinct* primes, $\mu(1) = 1$, and otherwise $\mu(n) = 0$, that is if $n$ contains *repeated* primes or if $n$ is not an integer. Euler's totient function $\phi(n)$ is defined for positive integers $n$ as the number of integers $n'$ such that $1 \leq n' \leq n$ and $n \wedge n' = 1$. Finally, $T_n(x)$ is the $n$'th order Chebyshev polynomial of the first kind; if we write $\sqrt{Q} = 2\cos(\pi e_0)$, then $T_{2d}\left(\tfrac{1}{2}\sqrt{Q}\right) = \cos(2\pi d e_0)$. With these definitions, the first few multiplicities read explicitly

$$\begin{aligned}
\alpha_{2,1}(Q) &= \tfrac{1}{2}(Q^2 - 3Q), \\
\alpha_{2,2}(Q) &= \tfrac{1}{2}(Q^2 - 3Q + 2), \\
\alpha_{3,1}(Q) &= \tfrac{1}{3}(Q^3 - 6Q^2 + 8Q - 3), \\
\alpha_{3,3}(Q) &= \tfrac{1}{3}(Q^3 - 6Q^2 + 8Q).
\end{aligned}$$

The important conclusion of this analysis is that the decomposition in (6) is *different* from the decomposition for the spin-1 anyonic chain. This means that different modules appear in the two models, and thus that their physical properties cannot be 'blindly' connected, even though they arise from the same underlying Temperley-Lieb algebra. Put differently, *we cannot straightforwardly obtain the properties of the anyonic model from the known properties of the Potts model.* A little extra work is needed, which we now carry out.

**Physics at $\theta_{2,1} = \theta_{\text{TL}} = -\pi/4$.** At this point the Hamiltonian is $H = -\sum E_i$. The value $k = 4$, that is $n = 2$, lies at the end of the corresponding Potts self-dual critical line, and the associated Hamiltonian is critical and described by a CFT with central charge $c = 1$. From the decompositions above, the Potts low-lying levels coincide with those of the $su(2)_4$ anyonic model (in particular their ground states are the same), so the latter is also described at this point by a $c = 1$ CFT . This is in agreement with the predictions of [9] that a continuous phase transition should occur at $\theta_{2,1} = -\pi/4$ for the $k = 4$ chain. For $k > 4$ however, that is $n > 2$, it is well-known that the corresponding Potts Hamiltonian (with $Q > 4$) is then at a point of *first-order phase transition*. The correlation length is *finite* [31], and the low-energy excitations are *not* described by a conformal field theory. Now the question is, are these properties essentially modified when going from the Potts representation to the anyonic representation?

For even sizes $L$, we find by exact study of small systems that the ground state of the anyonic chain is in fact the same as the ground state of the Potts model, and sits in the $\mathcal{V}_{0,1}$ module. This by itself shows that the anyonic chain cannot be described by a conformal field theory. Indeed, in a CFT, the ground state exhibits very special properties: for instance, the ground state energy has finite size corrections proportional to the central charge and the inverse of the chain length $L$, or the entanglement of a subsystem of size $l \ll L$ grows logarithmically with $l$ and is proportional to the same central charge [32]. None of these properties hold in the first-order transition point of the Potts model.[5] For odd sizes $L$, we find similarly that the ground state of the anyonic chain lies in the $\mathcal{V}_{1,\pi}$ module. This is not the ground state of the Potts model anymore. It is nonetheless a state lying at finite distance from it: the scaling of the corresponding energy or entanglement are thus similar, so low energy physics of the anyonic chain cannot be described by a CFT in this case either.

---

[5]This argument by itself leaves open the possibility that the central charge is zero. But this cannot be the case, for the anyonic model is unitary, and there are no unitary CFTs at $c = 0$ but the trivial one.

Let us now compare these conclusions with the results of [9]. For $k$ odd, the point $\theta_{\text{TL}}$ sits within the phase tentatively identified in [9] as a critical phase in the universality class of the minimal coset theory $su(2)_{k-1} \times su(2)_1/su(2)_k$. Clearly, this part of the phase diagram in [9] cannot be correct. For $k$ even, the point sits within what is described in [9] as an 'extended critical region', denoted in our figure 1 by a white region with a question mark. While for $k = 4$ this agrees with our finding that the point $\theta_{\text{TL}}$ is critical and described by a $c = 1$ CFT, for larger values of $k$ our analysis suggests that the situation is in fact more complicated, as this region supports at least one point not described by a CFT. We shall return to this point later, helped by further results from the neighboring integrable points.

The full algebraic analysis of which modules are relevant for which model would not really have been necessary to conclude that the anyonic chain is not at a conformal point for $\theta_{2,1} = \theta_{\text{TL}}$ - the information about the ground states was sufficient for this. There remains however the possibility that the anyonic chain may be gapless, with low energy excitations that are not described by a CFT but some other kind of field theory - eg one where space and time behave differently, so relativistic invariance is broken: it is to answer this question that the additional information about the spectrum is necessary. A full analysis of the excitations reveals a complex picture, with some gaps remaining finite as $L \to \infty$, while others are found to vanish like $1/L$. This is all however within the scope of what one can expect for a model at a first order transition point. Indeed, recent studies [33] have shown that, in the $Q > 4$ Potts model, depending on boundary conditions, gaps vanishing as $1/L$ in the thermodynamic limit could be observed, and we have observed gaps scaling in a similar way in the anyonic chain. The decomposition of the anyonic Hilbert space involves indeed modules $\mathcal{V}_{0,p}$. It is expected that the corresponding gaps vanish in the thermodynamic limit: the ground state is obtained for $p = 1$, and the effect of $p \neq 1$ is only a boundary effect, likely screened off quickly when the bulk correlation length is finite. In fact, by numerically solving the associated Bethe ansatz equations for sizes up to $L \sim 100$, we have found that the gaps between the ground state in $\mathcal{V}_{0,0}$ and those in $\mathcal{V}_{0,p}$ vanish like $1/L$ indeed (see figure 3). Other gaps like those between the ground state in $\mathcal{V}_{0,0}$ and the states in $\mathcal{V}_{2,0}$ are found to remain finite as $L \to \infty$ [6].

In conclusion, the anyonic chain is not quantum critical at $\theta_{2,1} = \theta_{\text{TL}}$ (and most likely sits at a first order transition point instead), and the $su(2)_{k-1} \times su(2)_1/su(2)_k$ phase for $k$ even proposed in [9] turns out to have more features than expected. It is possible that this fact was overlooked in [9] due to the fact that, for $Q > 4$ but not too large, the correlation length is known to be much larger than the sizes usually explored in simulations.

**Physics at $\theta_{2,1} = \pi + \theta_{\text{TL}} = 3\pi/4$** We now go back to the second Temperley-Lieb point, which corresponds to Hamiltonians $H = +\sum E_i$. The corresponding Potts model is less well studied, though a related loop model has been considered in [21, 22]. The corresponding RSOS model also appears to have been studied only sporadically; a few numerical results can be found in [22]. Concerning the range $n \geq 2$, we have strong reasons to believe that the work of [34] can be taken over, with only a few technical modifications, and that the conclusion will again be that there is a first-order phase transition. This fact agrees now with the predictions of [9], where the point $\theta_{2,1} = \pi + \theta_{\text{TL}} = 3\pi/4$ is found to be the locus of a first order phase transition separating an extended $Z_k$ parafermionic phase from the '$Z_3$-phase' described by the coset $su(2)_{k-4} \times su(2)_4/su(2)_k$.

---

[6]This discussion agrees with the observations of [31] that in order to create a finite gap in the massive six-vertex model one needs to make holes in the Fermi sea of Bethe integers, while only shifting these integers (which is the effect of a boundary twist) yields energies which are degenerate with the ground state in the thermodynamic limit.

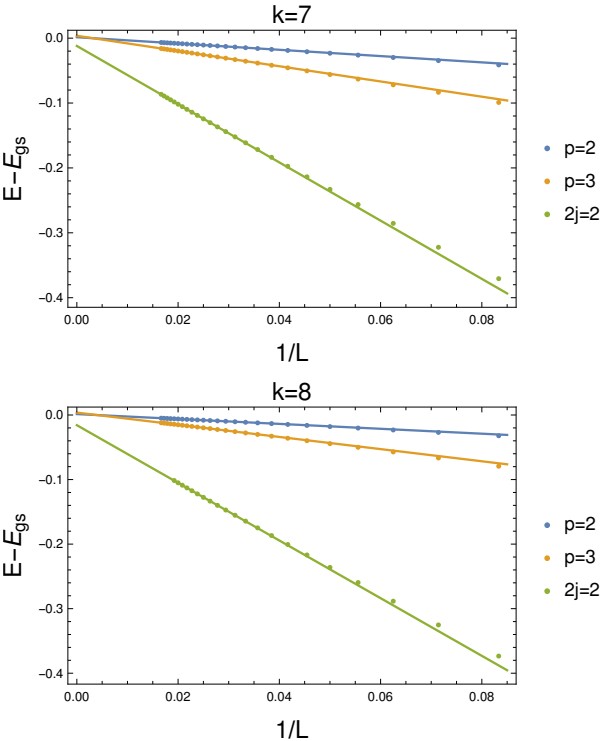

Figure 3: Gaps associated to the low-lying levels at the point $\theta_{\mathrm{TL}}$, obtained by numerical resolution of the Bethe ansatz equations, plotted as a function of $1/L$. The blue and orange dots correspond to gaps between the lowest lying levels in the $\mathcal{V}_{0,p}$ sector with $p = 2, 3$ and the ground state of the anyonic chain (lying in the $\mathcal{V}_{0,p=1}$ sector). The green dots correspond to a magnetic excitation, namely the lowest level of the $\mathcal{V}_{2j=2,0}$ sector. Solid lines correspond to fits of the form $E - E_{gs} = a + b/L$ of the largest size data. While for the magnetic excitation the gap remains finite in the $L \to \infty$ limit, for the $\mathcal{V}_{0,p}$ excitations it vanishes as $1/L$.

# 4 Integrability in spin-1 models

## 4.1 Spin-1 models

The core of the analysis in the foregoing section was to identify special points where the model of spin-1 anyons can be related with other models more naturally formulated in terms of spin-1/2. This is so because the Temperley-Lieb algebra appears most naturally in the context of spin-1/2 models: it is, for instance, a basic building block in the study of the golden chain [7], and, more formally, is identified as the centralizer of $U_q sl(2)$ in the product of spin-1/2 representations of the quantum group. There are other, more 'spin-1 like' integrable points in the phase diagram of the spin-1 anyonic chain. They are naturally related with representations of, this time, the Birman-Murakami-Wenzl algebra [35, 36]. The defining relations of this algebra in the context of our model are

$$B_i B_{i+1} B_i = B_{i+1} B_i B_{i+1},$$
$$\left[B_i, B_j\right] = 0, \quad \text{for } |i - j| \geq 2 \tag{9}$$

together with

$$E_i^2 = (1 + q^2 + q^{-2})E_i, \tag{10}$$
$$B_i E_i = E_i B_i = q^{-4} E_i,$$
$$E_i B_{i\pm1} E_i = q^4 E_i, \tag{11}$$

and

$$E_i E_{i\pm1} E_i = E_i,$$
$$B_i B_{i\pm1} E_i = E_{i\pm1} E_i. \tag{12}$$

It is easy to see that there exists a special combination of the anyonic projectors (2) that obeys these relations:

$$B = q^2 P^{(2)} - q^{-2} P^{(1)} + q^{-4} P^{(0)},$$
$$B^{-1} = q^{-2} P^{(2)} - q^2 P^{(1)} + q^4 P^{(0)},$$
$$E = (1 + q^2 + q^{-2}) P^{(0)}. \tag{13}$$

There are meanwhile two different solutions of the Yang-Baxter equation based on this algebra:

$$\check{R}_{\text{FZ}} \propto \text{Id} + \frac{x-1}{x+1} \frac{q^2 + x}{q^2 - x} E + \frac{1-x}{1+x} \frac{1}{q - q^{-1}} (B + B^{-1}),$$
$$\check{R}_{\text{IK}} \propto \text{Id} + \frac{x-1}{x+1} \frac{q^6 - x}{q^6 + x} E + \frac{1-x}{1+x} \frac{1}{q - q^{-1}} (B + B^{-1}). \tag{14}$$

It terms of the projectors in (2) these solutions read

$$\check{R}_{\text{FZ}} \propto P^{(2)} + \frac{q^4 x - 1}{q^4 - x} P^{(1)} + \frac{q^4 x - 1}{q^4 - x} \frac{q^2 x - 1}{q^2 - x} P^{(0)}, \tag{15}$$

$$\check{R}_{\text{IK}} \propto P^{(2)} + \frac{q^4 x - 1}{q^4 - x} P^{(1)} + \frac{q^6 x + 1}{q^6 + x} P^{(0)}. \tag{16}$$

The first solution is known as the Fateev-Zamolodchikov model [37], and is technically associated with $so(3)$ and its fundamental representation. The second solution is known as the Izergin-Korepin model [38], and is related with the twisted algebra $a_2^{(2)}$.

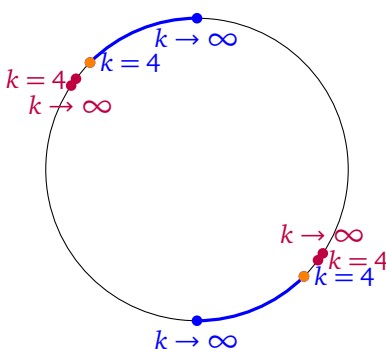

Figure 4: Location of the values of $\theta_{2,1}$ associated with the various integrable points considered in this paper, namely $\theta_{FZ}$ and $\pi + \theta_{FZ}$ (in red), $\theta_{IK}$ and $\pi + \theta_{IK}$ (in blue), $\theta_{TL}$ and $\pi + \theta_{TL}$ (in orange), as $k$ is varied from $k = 4$ to $k \to \infty$.

In these equations, $x = e^u$ where $u$ is the spectral parameter. Taking the derivatives of the $\check{R}$ matrices as $u \to 0$ gives integrable Hamiltonians. Going back to eq. (1) allows us to identify the corresponding values of the angle $\theta_{2,1}$:

$$\theta_{FZ} = -\arctan \frac{2\cos^2 \frac{\pi}{k+2}}{1 + 2\cos \frac{2\pi}{k+2}} \,, \qquad\qquad \pi + \theta_{FZ}$$

$$\theta_{IK} = -\arctan \frac{1}{\cos \frac{2\pi}{k+2} - \cos \frac{4\pi}{k+2}} \,, \qquad\qquad \pi + \theta_{IK} \,. \tag{17}$$

The location of these angles $\theta_{2,1}$ on the trigonometric circle when varying $k$ is summarized in figure 4. We note that the value $\theta_{FZ}$ was already identified in [9]. The others—especially $\theta_{IK}$ and $\theta_{IK} + \pi$—are new.

## 4.2 The Fateev-Zamolodchikov points

As mentioned earlier—as well as in [8, 9]—the anyonic models can be exactly mapped onto RSOS models. Such models with Boltzmann weights given by (15) have in fact been studied thoroughly in the integrable literature, where they are known as RSOS(2,2) models. In this case, they are tackled using 'fusion' of elementary RSOS models (or, equivalently, spin-1/2 anyonic chains) often denoted RSOS(1,1) [39–43]. Detailed studies using the Bethe ansatz as well as calculations of 'local height probabilities' have provided a complete understanding of these models, which are critical for both choices of $\theta_{2,1}$, but described by different universality classes [43].

### 4.2.1 The point $\theta_{FZ}$

The Hamiltonian corresponding to $\theta_{FZ}$ is associated with the so-called regimes III and IV in the denominations of [43], and is described in the continuum limit by the coset CFT

$$\frac{su(2)_{k-2} \times su(2)_2}{su(2)_k} \,,$$

with central charge

$$c = \frac{3}{2} - \frac{12}{k(k+2)} \,. \tag{18}$$

For $k > 4$, this coincides precisely with a critical point identified by [9] at the boundary of the Haldane phase, the so-called '$N = 1$ super CFT' point. For $k = 4$ the central charge is $c = 1$, and agrees once again with the predictions of [9].

### 4.2.2  The point $\pi + \theta_{\mathrm{FZ}}$

The Hamiltonian characterized by $\pi + \theta_{\mathrm{FZ}}$ is associated in the denominations of [43] with the so-called regimes I and II, and is described in the continuum limit by the $Z_k$ parafermionic CFT, which can also be formulated as the conformal coset

$$\frac{su(2)_k}{u(1)_{2k}},$$

with central charge

$$c = 2 - \frac{6}{k+2}. \tag{19}$$

This is in agreement with the predictions of [9], as we see from figures 1 and 4 that the point $\pi + \theta_{\mathrm{FZ}}$ lies in the extended parafermionic phase. Note that, as indicated in [9], this phase has a $Z_k$ sublattice symmetry, meaning that the true ground state, associated with the central charge (19), is only obtained for system sizes that are multiples of $k$.

## 5  Integrability and representations: the Izergin-Korepin points

Surprisingly, the RSOS models associated with the other solution of the Yang-Baxter equation have not, to our knowledge, been studied much in the literature. There is however a body of work on the related Izergin-Korepin vertex model [44–48]. This model is based on $R$-matrices and Hamiltonians that have exactly the form of (1) and (16) but with a different, non-anyonic Hilbert space, which is now the tensor product of spin 1 representations of $U_q sl(2)$, and gives rise to a spin-1 vertex model (with three states per site). Like in the Temperley-Lieb case before, the vertex model with twists is able to reproduce all modules of the Birman-Murakami-Wenzl algebra, and thus is the key, in principle, to the solution of all models based on this algebra. The vertex model having a rather complicated physics, the discussion is however more complex than in the Temperley-Lieb case, and we will not embark on a general study in terms of modules, but only discuss some crucial facts.

### 5.1  The point $\theta_{\mathrm{IK}}$

The vertex model relevant to our understanding of the anyonic chain at $\theta_{\mathrm{IK}}$ is known as the Izergin-Korepin or $a_2^{(2)}$ model in regime III [47]. Its continuum limit is extremely peculiar: it corresponds to a field theory on a non-compact target called 'Witten's black-hole CFT' [49,50], and which can be described technically as the $SL(2,\mathbb{R})/U(1)$ coset.

Let us recall the most important features relevant to our problem. The eigenstates of the periodic chain can be labeled in this regime by three integers: the total magnetization $m$, the momentum (eigenvalue of the translation operator) $w$, as well as some further integer index $j$ labeling the eigenstates within each sector of fixed $m, w$. The scaling of the corresponding energy levels with the system size ($L$) obeys the usual prediction of conformal field theory [51, 52]

$$E_{m,w,j}(L) = L E(\infty) - \frac{\pi v_f c_{m,w,j}}{6L} + \mathcal{O}(L^{-2}), \tag{20}$$

where $v_f$ is the Fermi velocity (a non-universal factor which depends on the normalization of the Hamiltonian), and $c_{m,w,j}$ is the so-called effective central charge associated the level

$(m, w, j)$. The ground state, corresponding to $(m, w, j) = (0, 0, 0)$, determines the true central charge $c$, while excited states are characterized by the conformal dimensions

$$\Delta_{m,w,j} + \bar{\Delta}_{m,w,j} = -\frac{1}{12}(c_{m,w,j} - c).$$

The spectrum of conformal weights was found in [47] to be made of one discrete part and one *continuous* part, a very unusual fact characteristic of the non-compact nature of the corresponding CFT. More precisely, focusing on the translationally-invariant, zero-momentum sector for conciseness, the conformal spectrum of the periodic chain was found to be of the form

$$-\frac{c_{m,0,j}}{12} = -\frac{2}{12} + \frac{m^2}{2(k+2)} + \left(N_{m,j}\right)^2 \frac{A(k)}{\left[B_{m,j}(k) + \log L\right]^2} \, . \tag{21}$$

Here, we have used the usual parameter $k$ to parametrize the quantum group deformation parameter $\mathfrak{q} = e^{\frac{i\pi}{k+2}}$. Recall that, while the anyonic and RSOS models can be defined only for $\mathfrak{q}$ a root of unity—and thus rational values of $k$—the vertex model can be defined for continuous values. The amplitudes $A(k)$ and $B(k)$ are continuous functions of $k$, and $N_{m,j}$ goes asymptotically as $2j + 1$. In the limit $L \to \infty$ the index $j$ can therefore be traded for a continuous index $s \sim j/\log L$: on top of the discrete part (quadratic in $m \in \mathbb{Z}$) of the set of exponents, we see that there is now a continous part, quadratic in $s$, with $s$ continuous. This can be related with the fact that the target (the space on which the fields take their values) of the associated field theory is non-compact. Of course, we do not expect the conformal field theory limit of the anyonic models (which are discrete and hermitian, hence corresponding to unitary CFTs) to exhibit a continuous spectrum of critical exponents. The mechanism relating the exponents of the anyonic chain to those of the vertex model is consequently rather subtle.

In order to recover the spectrum of the RSOS model (anyonic chain) from that of the vertex transfer matrix (Hamiltonian), one must consider twisted boundary conditions, parametrized by some twist angle $\varphi$ (which is set $= 0$ in the purely periodic case), just like we did in the Temperley-Lieb case earlier [7]. While this is quite innocuous for the finite-size spectrum (in particular it does not lead to any noticeable level crossing), this has now (unlike in the Temperley-Lieb case) some dramatic consequences on the conformal spectrum, as it leads to the appearance of *discrete states* popping out of the continuum at well-determined values of $\varphi$. More precisely, the effective central charges are determined by either one of two analytical formulae, depending on the value of $\varphi$, namely (restricting to $0 \leq \varphi \leq \pi$)

$$c_{m,0,j}(\varphi) = \begin{cases} c_m^* \equiv 2 - \frac{3}{2}(k+2)\left(\frac{\varphi}{\pi}\right)^2 - \frac{6m^2}{k+2} - \frac{48Aj^2}{(\log L)^2} & \text{for } \varphi \leq (|m| + 2j + 1)\frac{2\pi}{k+2} \\ c_m^* + \frac{6}{k}\left[\frac{k+2}{2}\left(\frac{\varphi}{\pi}\right) - (|m| + 2j + 1)\right]^2 & \text{for } \varphi \geq (|m| + 2j + 1)\frac{2\pi}{k+2}, \end{cases} \tag{22}$$

where in the first line we have used short-hand notations for the continous part of the spectrum. This discrete state mechanism is illustrated in figure 5.

From the exact diagonalization of the RSOS transfer matrix (anyonic chain Hamiltonian) for small finite system sizes (up to $L = 14$), we find that the lowest-lying levels are states of momentum $K = 0$ (including, in particular, the ground state) and $K = \pi$. Let us discuss these independently.

**Momentum $K = 0$ states.** From comparison of the anyonic and vertex spectra obtained from exact diagonalization of the corresponding Hamiltonians for system sizes $L = 4, \dots 14$,

---

[7]Note in particular that non-zero momentum states can be obtained from the zero momentum ones by increasing the twist by amounts of $\varphi$, so the above discussion of the vertex model eigenstates allows for full generality.

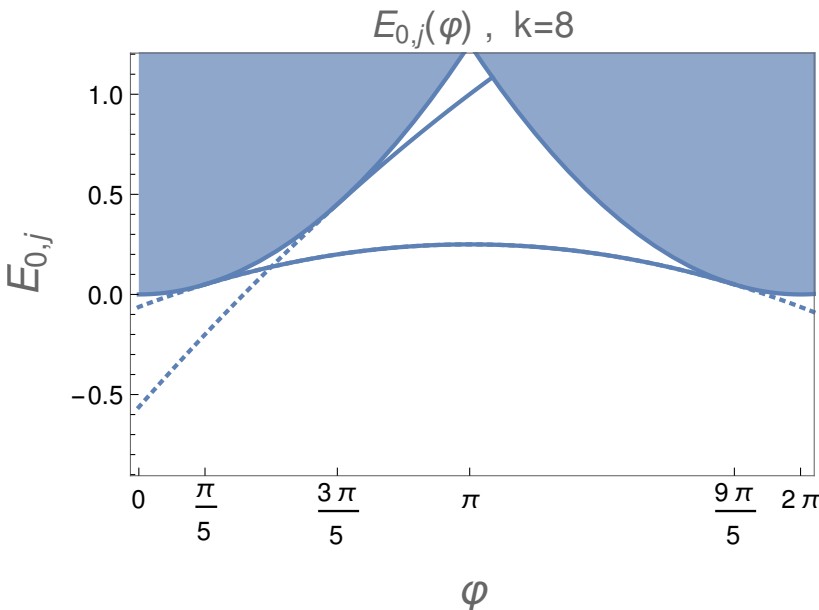

Figure 5: Discrete states structure of the Izergin-Korepin vertex model as a function of the boundary twist $\varphi$. We plotted on the vertical axis the rescaled energies of the levels $E_{0,j}$ (in the sector of magnetization $m = 0$), related to the central charges (22) by $E_{m,j} = -\frac{1}{12}(c_{m,0,j}-2)$ (therefore related to the conformal weights). The shaded area represents the continuum of states. Normalizable (resp. non-normalizable) discrete states present in (resp. absent from) the physical spectrum are represented by solid (resp. dashed) lines.

we observe that the low-lying levels of the former in the zero-momentum sector correspond to the following levels of the vertex model: we have state $(m, w, j) = (0,0,0)$ for twist values $\varphi = \frac{2\pi}{k+2}, \frac{4\pi}{k+2}, \frac{6\pi}{k+2}, \frac{8\pi}{k+2}, \ldots$, state $(0,0,1)$ for twist values $\varphi = \frac{6\pi}{k+2}, \frac{8\pi}{k+2}, \ldots$, etc... In other words, the zero momentum low-lying levels of the anyonic chains correspond to the states $(0,0,j)$ of the vertex model, for twist values $\varphi = \frac{2(2j+1)\pi}{k+2}, \frac{2(2j+2)\pi}{k+2}, \ldots$. Noticeably, all these states are in the discrete state regime, namely the twist $\varphi$ is sufficiently large for the second line of (22) to apply. The ground state $(m, w, j) = (0,0,0)$ is given by $\varphi = \frac{2\pi}{k+2}$—right at the intersection between the two expressions (22)—yielding the central charge

$$c_{\text{RSOS}} = c_{0,0,0}\left(\varphi = \frac{2\pi}{k+2}\right) = 2 - \frac{6}{k+2}, \qquad (23)$$

that is, the central charge of $Z_k$-parafermionic CFTs. This will be confirmed numerically, as shown in figure 6 and explained below in the text. The conformal weights associated with the excited states $(0,0,j)$, $\varphi = \frac{2\pi(2j+1+n)}{k+2}$, $j = 0,1,\ldots$, $n = 0,1,\ldots$ are further obtained as

$$\Delta = \bar{\Delta} = -\frac{1}{24}\left(c_{0,0,j}\left(\varphi = \frac{2\pi(2j+1+n)}{k+2}\right) - c_{\text{RSOS}}\right) = \frac{(2j+n)(2j+n+2)}{4(k+2)} - \frac{n^2}{4k}, \qquad (24)$$

which are precisely the dimensions of the parafermionic primary fields, parametrized in [9] in terms of the two integers $l = 2j + n$, $m = n$.

**Momentum $K = \pi$ states.** For even $k$, the low-lying states of the anyonic chain are found to correspond to the continuation to $\varphi = \pi, \pi - \frac{2\pi}{k+2}, \pi - \frac{4\pi}{k+2}, \ldots$ of the state of lowest energy level in the vertex model at $\varphi = \pi$. This state is however not the continuation of the ground

state $(0,0,0)$, as the two undergo a level crossing when the twist is varied from $0$ to $\pi$. The effective central charge associated with this state is found from numerical resolution of the BAE to be of the form

$$c_{\text{eff}} = \frac{1}{2} - \frac{3(k+2)}{2}\left(\frac{\varphi}{\pi}\right)^2 , \qquad (25)$$

so the exponent associated to $\varphi = \pi - p\frac{2\pi}{k+2}$ (with $p = 0, 1, \ldots, \frac{k}{2}$) reads

$$\Delta + \bar{\Delta} = \frac{1}{8} + \frac{p^2 - 1}{2(k+2)} . \qquad (26)$$

Setting $r = \frac{k+2}{2} - p$, and using the fact that the states have zero conformal spin (so $\Delta = \bar{\Delta}$), we get

$$\Delta = \bar{\Delta} = \frac{k - 2 + (k+2-2r)^2}{16(k+2)} , \qquad r = 1, 2, \ldots, \frac{k+2}{2} , \qquad (27)$$

which is precisely [53, eq. (2.11)] for the $C$-disorder fields in the $Z_k$ parafermionic theory.

For odd $k$, it is more difficult to understand the correspondence between the leading eigenvalues in the $K = \pi$ sector and the levels of the RSOS model. However, we find numerically that those can also be described by (26), with this time $p = \frac{1}{2}, \frac{3}{2}, \ldots, \frac{k}{2}$. This will be confirmed by the analysis displayed in figure 7 (see the following paragraph). Using once again the parametrization $r = \frac{k+2}{2} - p$ the exponents are again those of the $C$-disorder fields, with $r = 1, 2, \ldots, \frac{k+1}{2}$. This again is precisely the interval given in [53].

**Numerical checks.** We now present a numerical check of the above results. In [9], the central charge and conformal weights at critical points or in extended critical regions were measured from the finite-size energies through the usual scaling formulas predicted by CFT [51,52]

$$E(L) = LE(\infty) - \frac{\pi v_f}{6L}(c - 12(\Delta + \bar{\Delta})) + \mathcal{O}(L^{-2}) ;$$

see also eq. (20) above. The extensive energy $E(\infty)$ and Fermi velocity $v_f$ are generically unknown, and in [9] the latter is adjusted numerically so as to match a consistent CFT description. In the present integrable case, however, both $E(\infty)$ and $v_f$ can be determined analytically through Bethe ansatz calculations [47]. These are originally computed for the vertex model (spin chain), but do not depend on the boundary twist and therefore also hold for the anyonic chain. Taking for the spin chain Hamiltonian

$$H = \sum_i \left( -\tan\frac{3\pi}{k+2} P_i^{(0)} + \frac{1}{\tan\frac{2\pi}{k+2}} P_i^{(1)} \right) , \qquad (28)$$

which is related to $H_{\text{IK}} = \sum_i \left( \cos\theta_{\text{IK}} P_i^{(2)} - \sin\theta_{\text{IK}} P_i^{(1)} \right)$ by some positive multiplicative factor and identity shift, we find

$$E(\infty) = \int_{-\infty}^{\infty} du \, \frac{2}{\pi} \frac{\sin\left(\frac{2\pi}{k+2}\right)\frac{k+2}{k-4}}{\cosh\left(2u\frac{k+2}{k-4}\right)} \frac{\cosh 2u \sin\frac{\pi}{k+2} - \cos\frac{2\pi}{k+2}}{\cos\left(\frac{2\pi}{k+2}\right)^2 + \frac{1}{2}\left(\cosh 4u - \cos\frac{2\pi}{k+2}\right) - 2\cos\frac{2\pi}{k+2}\sin\frac{\pi}{k+2}\cosh 2u}$$
$$v_f = \frac{k+2}{k-4} . \qquad (29)$$

Using these expressions already removes a lot of indetermination from the numerical fits. In figure 6, we measure the associated central charge, whose convergence towards the parafermionic value (23) is clear. In figure 7, we display the structure of the excitations spectrum for $k = 5$ and $k = 6$, in the fashion of the figures given in [9]. At low energy the presence of the conformal weights given by (24) and (26) in the spectrum is manifest.

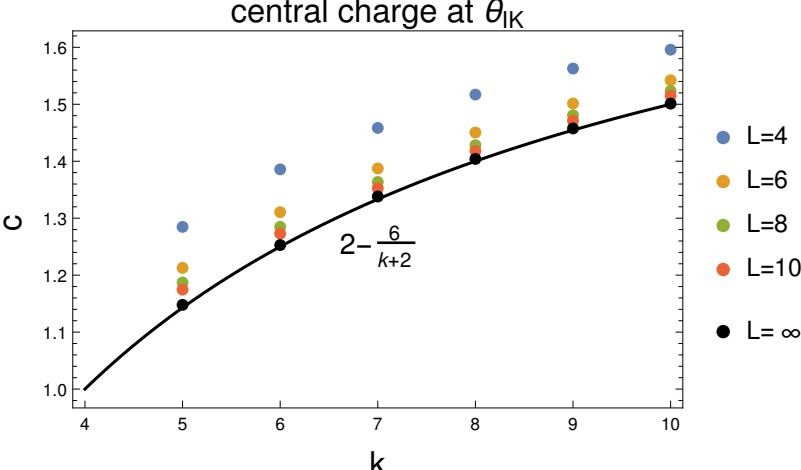

Figure 6: Central charge at the point $\theta_{\mathrm{IK}}$, obtained from exact diagonalization of the Hamiltonian (28) for finite system sizes, and using the expressions (29) of the thermodynamic limit free energy and Fermi velocity. The black dots are an extrapolation to $L \to \infty$ using a quadratic fit in $L^{-1}$. In comparison, we plotted in black the parafermionic central charge (23).

**Comparison with the extended parafermionic phase.** From the above analysis we conclude that the continuum limit of the spin-1 $su(2)_k$ anyonic chain at the point $\theta_{\mathrm{IK}}$ is described by the $Z_k$ parafermionic theory. For $k > 4$, this result disagrees with the phase diagram proposed in [9]: for $k$ odd, the point $\theta_{\mathrm{IK}}$ indeed lies in the region believed in [9] to be described by the $\frac{su(2)_{k-1} \times su(2)_1}{su(2)_k}$ coset (while for $k$ even no conjecture is made for the corresponding CFT).

Noticeably, this critical point seems of a different nature than the other, extended parafermionic phase proposed in [9], as can be seen from the comparison of the conformal spectrum of our figure 7 with that observed in [9]. Among the notable differences, we point out:

1. While the extended $Z_k$ phase was observed to have an underlying $Z_k$ sublattice symmetry, such a symmetry is absent at $\theta_{\mathrm{IK}}$, namely the true ground state leading to the parafermionic central charge is present for any value of the system size $L$.

2. There are extra conformal weights (27) at the $\theta_{\mathrm{IK}}$ point, corresponding to the $C$-disorder fields studied in [53].

3. The absence of topological protection: Eigenstates of the anyonic chain can be classified according to a topological number, which in the parafermionic CFT was identified [8, 9] as $l \equiv 2j + n \pmod{k}$. In the extended $Z_k$ phase, it was observed in [8] that no relevant field with the same topological dimension as the identity, namely $2j + n = 0$, is present in the $K = 0$ sector. Resultingly, the $Z_k$ behaviour was deduced to be *topologically protected*, namely it cannot be driven off criticality by a perturbation preserving both the translational invariance and topological symmetry.

   This is not true anymore at the $\theta_{\mathrm{IK}}$ point, where we see that many more fields are present in the $K = 0$ sector. This is the case, in particular, of the field with labels $(j, n) = (1, k-2)$, which has a topological index $2j + n = k \equiv 0$, and is the most relevant such field (apart from the identity), with dimension

$$\Delta + \bar{\Delta} = 2\frac{k-1}{k}. \tag{30}$$

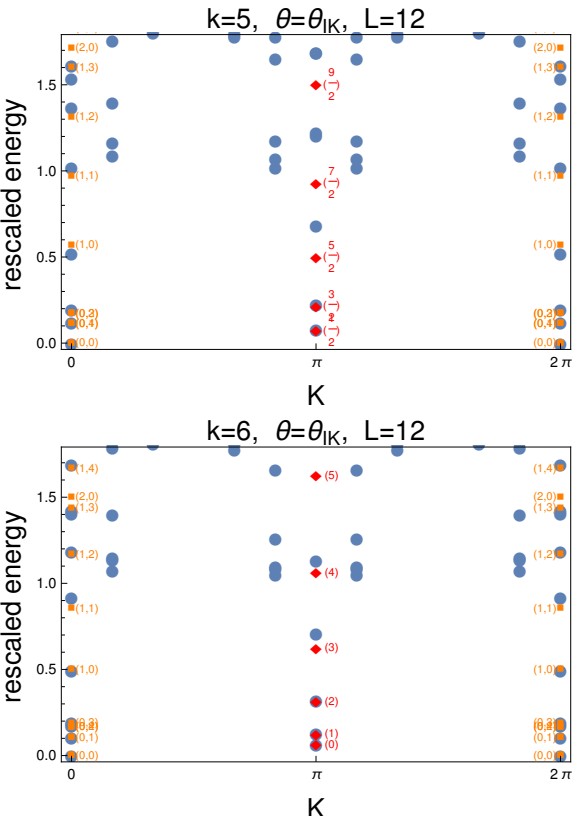

Figure 7: Spectrum of the $k = 5$ and $k = 6$ chains with $L = 12$ sites at the point $\theta_{2,1} = \theta_{\text{IK}}$, rescaled such that the vertical axis measures the scaling dimensions $\Delta + \bar{\Delta}$. The eigenvalues are classified according to the momentum $K = 0, \frac{2\pi}{L}, \frac{4\pi}{L}, \ldots, \frac{2\pi(L-1)}{L}$, and are displayed as blue dots. In comparison, we indicate by orange (resp. red) squares the predictions of eq. (24) (resp. eq. (26)), labeled by the two integers $(j, n)$ (resp. $p$). The corresponding topological indices are for the former given by $l = 2j + n \pmod{k}$ and indicated in black, while for the latter they are zero. While the convergence at this size gets poorer for higher excitations, the lowest-lying levels are in great agreement with the analytical expectations.

Consequently, we should expect that this field drives a transition across $\theta_{\text{IK}}$, or, in other terms, that $\theta_{\text{IK}}$ does not belong to an extended phase. We will examine this issue in further detail in section 6.

## 5.2  The point $\pi + \theta_{\text{IK}}$

The Hamiltonian associated with $\pi + \theta_{\text{IK}}$ can similarly be understood from the analysis of the $a_2^{(2)}$ vertex model. In its vertex formulation, the Hamiltonian is described by the so-called regime I [46], whose continuum limit is that of a compactified free bosonic field. However, we find that the corresponding eigenlevels are absent from the RSOS version, providing another illustration of the fact pointed out earlier that the physics can very much depend on the representation. Instead, the lowest-lying eigenlevels of the RSOS Hamiltonian are those of what two of the present authors have called 'regime IV' in another recent work [23]. There, the corresponding physics (which turned out to be related to that of a colouring problem on the triangular lattice) was understood from a detailed Bethe ansatz analysis, and the continuum limit described as the coset CFT

$$\frac{su(2)_{k-4} \times su(2)_4}{su(2)_k}, \tag{31}$$

in agreement with the phase diagram of [9].

# 6  Conclusion

A careful analysis of the integrable points of the spin-1 anyonic chain has revealed some surprises. To a considerable extent, the points $\theta_{\text{TL}}$, $\theta_{\text{IK}}$ have a physics which is **different** (for $k > 4$) from what was expected from the analysis in [9]. The point $\theta_{\text{TL}}$ is a point of first-order phase transition and the point $\theta_{\text{IK}}$ is in the universality class of $Z_k$ parafermions, while in [9] these two points were misidentified as belonging to a critical phase in the universality class of $su(2)_{k-1} \times su(2)_1/su(2)_k$ coset models for $k$ odd. The situation for $k$ even is a bit more confusing. The point $\theta_{\text{TL}}$ sits in the middle of a phase (between "dimerized" and "Haldane") that was not fully identified in [9]: we now know that it is a point of first-order phase transition. As for the point $\theta_{\text{IK}}$ for $k$ even, we will show in the following that it is located at the boundary of the dimerized phase, whose properties were left unidentified in [9] (cf. the point marked by a red question mark in figure 1).

We can now, with these new elements in hand, derive some aspects of the modified phase diagram of the model. Before doing so, however, it is useful to have some extra information about the vicinity of our new points $\theta_{\text{IK}}$ with $Z_k$ continuum limit.

## 6.1  $Z_k$ perturbations at $\theta_{\text{IK}}$

At the end of section 5.2 we have shown that the operator that is associated with the labels $(j, n) = (1, k-2)$, of topological index 0, is present in the $K = 0$ sector of the Hamiltonian at $\theta_{\text{IK}}$, and should therefore drive a transition across this critical point.

In the $Z_k$ parafermionic corformal field theory, the so-called parafermion currents $\psi_r(z)$, with $r = 1, 2, \ldots, r-1$, are defined. They have dimension $\Delta_r = \frac{r(k-r)}{k}$, and a charge conjugation defined by $\psi_r^+ = \psi_{k-r}$. Using the result (24) these fields can be identified with the fields $(j, n)$, where $j = r$ and $n = k - 2r \pmod{n}$, so in particular the field $(j, n) = (1, k-2)$ can be identified with $\psi_1$, or $\psi_1^+ = \psi_k$. The corresponding perturbation of the parafermionic CFT is precisely the one studied in [54], with action

$$\mathscr{A} = \mathscr{A}_{Z_k} - \lambda \int \mathrm{d}^2 x \left( \psi \bar{\psi} + \psi^+ \bar{\psi}^+ \right). \tag{32}$$

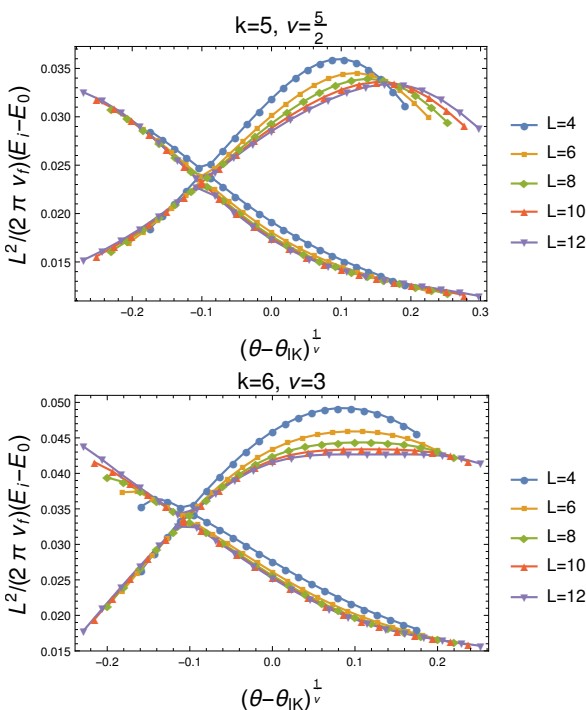

**Figure 8:** Rescaled gaps $\frac{L^2}{2\pi v_f}(E_i - E_0)$ associated with the two first excited levels, $E_1$ and $E_2$, of the $k = 5, 6$ anyonic chains in the vicinity of the point $\theta_{\mathrm{IK}}$, plotted against $\left(\theta_{2,1} - \theta_{\mathrm{IK}}\right) L^{\frac{1}{\nu}}$ for $\nu = \frac{1}{2-x} = \frac{k}{2}$. The scaling collapse is quite convincing.

This perturbed CFT is known to flow for $\lambda$ negative towards a critical phase described by the minimal model $\mathcal{M}_{k+1}$, namely the coset $\frac{su(2)_{k-1} \times su(2)_1}{su(2)_k}$ with central charge

$$c = 1 - \frac{6}{(k+1)(k+2)}, \tag{33}$$

and for $\lambda$ positive towards a massive phase. This suggests that the perturbation (32) describes the continuum analog of $\theta_{2,1}$ perturbations around $\theta_{\mathrm{IK}}$, namely $\theta_{2,1} - \theta_{\mathrm{IK}} \propto \lambda$. For $\theta_{\mathrm{IK}}$ we indeed recover the $\frac{su(2)_{k-1} \times su(2)_1}{su(2)_k}$ theory observed in the phase diagram, and we are therefore led to speculate that for $\theta_{2,1} > \theta_{\mathrm{IK}}$ there should be a massive phase.

We can study the behaviour of the scaled gap $x_t = \frac{L^2}{2\pi v_f}(E_1 - E_0)$ between the leading eigenvalues of the anyonic chain Hamiltonian (where we take for $v_f$ the value analytically derived at the $Z_k$ point). Around a critical point perturbed by a relevant operator $x = \Delta + \bar{\Delta}$, the scaled gap is expected to scale as $x_t \sim F\left(\left(\theta_{2,1} - \theta_{\mathrm{IK}}\right) L^{\frac{1}{\nu}}\right)$, where $\nu = \frac{1}{2-x} = \frac{k}{2}$ in the present case. We refer to figure 8, where the gaps associated with the two leading excitations are plotted against $\left(\theta_{2,1} - \theta_{\mathrm{IK}}\right) L^{\frac{1}{\nu}}$. The two excited states undergo a level crossing for $\theta_{2,1}$ slightly smaller than $\theta_{\mathrm{IK}}$, and we focus on the gap obtained by following analytically the leading excitation at $\theta_{\mathrm{IK}}$. For this gap the collapse between data for different system sizes suggests that the value of $\nu$ is the correct one.

To end this discussion, we briefly consider the limit $k \to \infty$ limit where $\Delta_r + \bar{\Delta}_r = 2$ and the perturbation (32) becomes exactly marginal. Formally, the limit $k \to \infty$ of the model is described by two non-compact bosons, and, if we extrapolate what we know about the massive and massless flows emanating from the $Z_k$ theories, should correspond to the UV limit of the $O(3)$ sigma model (at $\theta = 0$ and $\pi$ respectively) [54]. However, it is not clear what happens

to the lattice model, and the scaling limit and the $k \to \infty$ limits do not seem to commute. A similar situation was encountered in [55].

## 6.2 The modified phase diagram

The foregoing discussion agrees with the idea that the $su(2)_{k-1} \times su(2)_1 / su(2)_k$ phase terminates at $\theta_{\text{IK}}$ for $k$ odd. Since meanwhile the Haldane phase for $k$ odd terminates at $\theta_{\text{FZ}}$, this suggests the existence of at least one new phase between these two values of $\theta_{2,1}$ for $k$ odd, of which we know very little, except that it contains a point of first-order phase transition at $\theta_{\text{TL}}$.

Things for $k$ even are much less clear. Numerical calculations do not show a clear distinction between $k$ odd and $k$ even near the $\theta_{\text{IK}}$ point. On the other hand, it is difficult to see how one could have a massless phase with $su(2)_{k-1} \times su(2)_1 / su(2)_k$ symmetry bordering this point. Most likely, the RG flows that occur for $k$ odd are prevented by less relevant terms that take the theory to other, essentially trivial fixed points. This does not help characterizing the phase between the Haldane and dimerized phase much.

## 6.3 A look at $k = 6$

We finally consider the case $k = 6$. Since the fusion rules of $su(2)_k$ are symmetric under the exchange $j \leftrightarrow \frac{k}{2} - j$, the spin-1 and spin-2 channels are equivalent in this case, so the set of energy eigenlevels is symmetric under $\theta_{2,1} \leftrightarrow \frac{3\pi}{2} - \theta_{2,1}$, namely a reflexion whose fixed points are $\theta_{\text{TL}} = \frac{3\pi}{4}$ and $\pi + \theta_{\text{TL}} = -\frac{\pi}{4} \pmod{2\pi}$. As underlined in [9], we point out that only the set of energy levels is symmetric, but not necessarily the corresponding eigenvalues, neither the momentum sectors in which these eigenvalues lie. For instance the massive 'Haldane' and dimerized phases are symmetric of each other; but while in the former there is a twofold ground state degeneracy (exact only at the AKLT point $\theta_{2,1} = 0$, but recovered in the continuum limit in the whole phase) between two states of zero momentum, in the latter a similar degeneracy (exact only at $\theta_{2,1} = \frac{3\pi}{2}$) holds between one state of momentum 0, and one state of momentum $\pi$.

At $k = 6$, the different integrable points are located as follows: $\theta_{\text{TL}} = \frac{3\pi}{4}$, $\theta_{\text{IK}} = -\arctan\sqrt{2}$, and $\theta_{\text{FZ}} = -\arctan\frac{1}{\sqrt{2}}$. So in particular $\theta_{\text{FZ}}$ and $\theta_{\text{IK}}$ are related by symmetry, and since the former was identified in section 5 as the boundary of the Haldane phase, the latter should similarly correspond to the boundary of the dimerized phase. Moreover, the two corresponding CFTs should coincide, which is indeed the case, since for $k = 6$ the coset $\frac{su(2)_{k-4} \times su(2)_4}{su(2)_k}$ is equivalent to the $Z_6$ parafermionic theory.

## 6.4 Conclusions about the phase diagram

Our results bring new insights to the study of the phase diagram of the $su(2)_k$ spin-1 anyonic chain. While for $k = 4$ all our results coincide with the predictions of [9], several novelties appear for $k > 4$. We see in particular that the cases of $k$ odd and $k$ even are much closer than initially believed. Some key questions however remain unanswered. In particular:

- What is the nature of the conjectured massive phase for $\theta_{2,1} > \theta_{\text{IK}}$?

- What is on the other side on the first-order transition at $\theta_{\text{TL}}$, namely between $\theta_{\text{TL}}$ and $\theta_{\text{FZ}}$? Can we understand it as a perturbation from the supersymmetric CFT at $\theta_{\text{FZ}}$?

Our findings can be summarized in figure 2.

## Acknowledgements

EV acknowledges discussions with P. Fendley, E. Ardonne, P.A. Pearce and A. Klümper.

**Funding information** The work of EV was supported by the ERC under Starting Grant 279391 EDEQS. The work of HS and JLJ was supported by the ERC Advanced Grant NuQFT. The work of HS was also supported by the US Department of Energy (grant number DE-FG03-01ER45908), and the work of JLJ was also supported by the Institut Universitaire de France.

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
