# Peer review of "Elaborating the phase diagram of spin-1 anyonic chains"

_SciPost Physics, doi:SciPost Phys. 2, 004 (2017)_

## Round 1 · Referee Report · Anonymous (Referee 1) · 2016-11-9

Strengths

  1. clear presentation
  2. clear discussion of relation to literature on anyonic chain
  3. clear discussion of special cases

Weaknesses

  1. first paragraph of introduction not very informative
  2. in Sec. 2 it is stated that $k\ge 2$ is assumed. However, the used fusion rules $1\times 1=0+1+2$ suggest that $k\ge 4$ is intended.
  3. The description in Fig. 2 also requires $k\ge 4$.
  4. several references are incomplete, eg, in Ref 7 the page is missing
  5. page number is missing in the reference stated In the abstract
  6. for the discussion of $su(2)_k$ it is referred to Refs 7,8, so the paper is not self contained (and $su(2)_k$ is not text book material)

Report

The authors study several integrable points in the spin-1 anyonic chain and use them to obtain information about the general phase diagram. They discuss in particular the relation of their findings to the original reference (Ref 8). Overall the quality of the presentation is very good.

Requested changes

  1. Correct references.
  2. Clarify the restriction on $k$.
  3. If really $k\ge 2$ is intended, then add a discussion of the special cases $k=2$ and $k=3$.

---

## Round 1 · Referee Report · Anonymous (Referee 2) · 2016-11-18

Strengths

1- povides a comprehensive overview on integrable spin-1 chains (vertex models) related to the $su(2)_k$ anyon chains 2- conclusions on CFTs are supported by numerical work (mostly for $k=5,5$)

Weaknesses

1- although not stated clearly, the phase diagrams and most of the results appear to be for $k>4$ only. 2- in fact, the $su(2)_4$ model is integrable for any coupling and known to be in the Ashkin-Teller class of models (see work on magnetic hard squares by Pearce&Kim (JPhys A20, 6471, '87) and, more recently, in Ref.[14] of the manuscript)

Report

The authors identify several integrable points among the spin-1 $su(2)_k$ anyon models introduced and studied numerically earlier. Their exact results are highly illuminating and allow for a significant refinement of the phase diagrams of the $k>4$ models as compared to Ref. [8].

Requested changes

1- the range of $k$ for which results hold should be stated clearly 2- known properties of the $k=4$ model should be referred to

---

## Round 1 · Referee Report · Anonymous (Referee 3) · 2016-12-19

Strengths

.

Weaknesses

.

Report

The manuscript at hand is a technical addendum to the extensive study of the physics of spin-1 anyonic chains presented by Gils and collaborators in a paper some few years ago (Ref. [8] in the manuscript at hand). Rightfully, the authors point out that this earlier study has missed to identify two exactly solvable “Izergin-Korepin” points and elaborate that the originally determined phase diagram of Gils et al. needs to be clarified. This is, of course, an interesting and valid result that merits publication in some form. I therefore endorse publication of the manuscript at hand in SciPost.

Before the paper should be published the authors should take care of the following aspects of their representation:

  • To say that “non Abelian anyons … have showed up” in a number of physical systems is a rather sloppy and incorrect wording for a paper that later on is arguing in rather rigorous (and sometimes dismissive) terms. Non-Abelian anyons are theoretically known to appear in px + i py superconductors, but have not been seen experimentally in this context. They have been proposed by Read and Green to be relevant to certain quantum Hall states, again with no experimental verification up to date. And whether the nanowires have seen non-Abelian end states should also be worded in a more tempered way.

  • One should generally distinguish “Majorana fermions” from “Ising anyons”. In the present context where representation of su(2)_k are studied, one should consistently choose the latter.

  • The first time the physics of a spin-1 anyonic chain has been discussed was in the context of Fibonacci anyons in Phys. Rev. Lett. 101, 050401 (2008), where e.g. the anyonic variant of the Haldane phase has been discussed. This reference should be included in the current discussion.

  • It would be instructive to contrast the earlier phase diagrams, now shown in Fig. 1, with the results of the current study, shown in Fig. 8, side-by-side at the beginning of the paper, e.g. in a new figure 1.

  • Figure 2 shows some numerical finite-size data with some finite-size extrapolation. However, in the current form this finite-size extrapolation appears to be dominated by the smallest system sizes with a rather noticeable discrepancy for the largest system sizes. There is no need to fit all the finite-size data, in fact it would be much more appropriate to only fit the largest system sizes for the “2j=2” data sets. The extrapolated gap would be noticeably smaller. Is this still sufficient numerical evidence for a gapped phase?

  • In Figure 3 the authors use red and green labels, which are confusing colors for many people. Please replace one of the two colors.

  • The section headings 4.2.1 and 4.2.2 needs to be adjusted.

  • In equation (30) the scaling dimension of the relevant field is given. It would be interesting to discuss the k \to \infty limit here, where the operator becomes marginal. Can this su(2) limit be understood on general grounds?

  • In Figure 6 it would be informative to also label the indicated field by their topological sector.

Requested changes

.

---

## Round 2 · Author Response

We thank the referees for the constructive comments brought to our manuscript.
In the present version (v2), corrections were made in order to take in account of all received remarks.

In particular, the following changes were made :

---

## Round 2 · List of Changes

- Regarding the remark of Report 53 that
{\it
It would be instructive to contrast the earlier phase diagrams, now shown in Fig. 1, with the results of the current study, shown in Fig. 8, side-by-side at the beginning of the paper, e.g. in a new figure 1.}

We agree with this comment, however space and readability constraints made very inconvenient to put figures 1 and 8 side by side. We have instead moved fig. 8 to the beginning of the paper, where it can more easily be compared with fig. 1.

- Regarding the remark of Report 53 that
{\it
Figure 2 shows some numerical finite-size data with some finite-size extrapolation. However, in the current form this finite-size extrapolation appears to be dominated by the *smallest* system sizes with a rather noticeable discrepancy for the largest system sizes. There is no need to fit *all* the finite-size data, in fact it would be much more appropriate to only fit the largest system sizes for the “2j=2” data sets. The extrapolated gap would be noticeably smaller. Is this still sufficient numerical evidence for a gapped phase? }

We have replaced the extrapolations on figure 3 (prevously fig. 2) by linear fits excluding the smallest system sizes. The extrapolated gap is indeed smaller, but still manifestly non-zero. This is in agreement with the fact that magnetic excitations (corresponding to holes in the Fermi sea) of the six-vertex model in its massive phase have a non-zero gap.

- Regarding the remark of Report 53 that
{\it In Figure 6 it would be informative to also label the indicated field by their topological sector.}

We have added on the legend of Figure 7 (previously fig. 6) information about the topological charge of all indicated levels

- In figure 4 (formerly figure 3), we have changed the green color to orange, in order to improve its readability.

- Regarding remarks of Reports 32 and 37, we have specified the range of values of $k$ for which our results hold. More precisely, most of our conclusions are formulated for generic $k\geq 4$, but when necessary we have treated separately the cases $k>4$ and $k=4$.
In addition, we have refered to the suggested litterature for the $k=4$ case, for which we thank the author of Report 37.

- We have also improved the introduction in order to meet the suggestions of Report 53

You are currently on this page

Resubmission 1611.02236v2 on 9 February 2017

---

## Editorial Decision

published